# Immune Response in Regard to Hypersensitivity Reactions after COVID-19 Vaccination

**DOI:** 10.3390/biomedicines10071641

**Published:** 2022-07-08

**Authors:** Ming-Hsiu Hsieh, Yukie Yamaguchi

**Affiliations:** Department of Environmental Immuno-Dermatology, Yokohama City University Graduate School of Medicine, Yokohama 236-0004, Japan; yui1783@yokohama-cu.ac.jp

**Keywords:** COVID-19, vaccination, Pfizer, BioNTech, BNT162b2, Moderna, mRNA-1273, AstraZeneca, AZD1222, hypersensitivity

## Abstract

Severe acute respiratory syndrome coronavirus 2 (SARS-CoV-2), the virus that causes coronavirus disease 2019 (COVID-19), is a member of the genus *B**etacoronavirus*. This virus was first detected in December 2019, and the situation quickly escalated to cause a global pandemic within a few months. COVID-19 had caused more than 5.5 million deaths as of January 2022. Hence, the urgency of effective vaccination contributed to the fastest rate of vaccine development seen to date (i.e., within 1.5 years). Despite reports of good vaccine efficacy without severe systemic reactions at the clinical trial stage, hypersensitivity reactions have been reported following worldwide vaccination campaigns. We provide a brief review regarding the structure of SARS-CoV-2. We also review the most acceptable types of vaccines in terms of safety profiles, namely the BNT162b2, mRNA-1273, and AZD1222 vaccines. This review aims to facilitate an understanding of the possible immune mechanisms regarding COVID-19-vaccination-related hypersensitivity reactions, such as thrombosis and thrombocytopenia, cutaneous adverse reactions, myocarditis, and perimyocarditis.

## 1. Introduction

In 2019, a novel coronavirus disease, COVID-19, caused by severe acute respiratory syndrome coronavirus 2 (SARS-CoV-2), led to a global disaster. Among other devastating impacts, this disease had, till January 2022, led to more than 5.5 million deaths and has caused a profoundly detrimental economic impact due to the suspension of trade and quarantine policies between countries that have been in place since the end of 2019 [1].

COVID-19 was first reported in late December of 2019 in Wuhan, China. The clinical manifestations of this disease range from an asymptomatic flu-like syndrome to anosmia, atypical pneumonia, coagulopathy problems, and cytokine storms inducing multiple organ failure, and even death. In addition to the respiratory tract, patients frequently manifest symptoms in their gastrointestinal system, liver, heart, skin, and/or central nervous system [2,3]. In addition, there have been reports of hypersensitivity reactions following COVID-19 infection, including urticaria, morbilliform rash, pernio-like acral lesions, vasculitis, pemphigus vulgaris, and Stevens–Johnson syndrome [4,5,6]. The World Health Organization (WHO) termed this novel disease as “coronavirus disease 2019 (i.e., COVID-19)” on 11 February 2019, and declared the disease a global pandemic on 11 March 2020 [7].

Coronaviruses are positive-sense single-stranded RNA viruses (+ssRNA), named after their crown-like appearance due to spike glycoproteins on the spherical or pleomorphic envelope (Figure 1a) [2,3]. COVID-19 belongs to the beta genera of the *Orthocoronavirinae* subfamily of the *Coronaviridae* family, similar to the severe acute respiratory syndrome coronavirus (SARS-CoV) and the Middle East respiratory syndrome coronavirus [2,3]. Its origins have been traced to bats, and the virus was newly transmitted to humans without a clear underlying reason.

A previous genomic characterization from a patient with COVID-19 revealed 89% and 82% nucleotide identity with bat SARS-like-CoVZXC21 and human SARS-CoV, respectively. Therefore, the International Committee on Taxonomy of Viruses termed this the novel virus SARS-CoV-2 [2,7,8]. This virus gradually developed mutations, which have been identified as Alpha, Beta, Gamma, Delta, and Omicron variants according to the most recent communications from the WHO [9].

The WHO has defined a successful vaccine as one with >30% vaccine efficacy and at least 50% disease reduction [3]. Traditionally, vaccine development (from the exploratory and pre-clinical stage to approval and surveillance) requires approximately 10–15 years and a budget of approximately USD 1 billion [10,11]. The following steps are usually performed before vaccine manufacturing and post-marking surveillance: (1) an exploratory and pre-clinical stage for antigen identification, cell or tissue culture, and/or animal trials in order to determine the vaccine’s safety, immunogenicity, and efficacy; (2) phase I, II, and III clinical trials starting from small groups of healthy individuals, continuing to hundreds of individuals classified into different demographic groups, and eventually continuing into large-scale trials of thousands of individuals for the purpose of determining vaccine safety, efficacy, and appropriate dosages; (3) review and licensing by the U.S. Food and Drug Administration (FDA), the WHO, or the European Medicines Agency; and (4) manufacturing, phase IV clinical trials, and post-marking surveillance in order to assess vaccine effectiveness as well as adverse effects following vaccine approval [3,10,11]. Each stage and trial generally takes two–three years to complete.

However, considering the grim urgency in the wake of the COVID-19 pandemic, the vaccine development phases were combined to form as phase I plus II trials in hundreds of people, and the approval process has often been simplified in order to greatly decrease the total period of vaccine development [10]. Despite these simplified development and approval procedures, several gold-standard clinical trials have revealed good vaccine efficacy without severe and/or systemic reactions (aside from local injection reactions, fever, and fatigue) with a high standard of evidence [12,13]. However, following worldwide vaccination campaigns, some people have reported unexpected hypersensitivity reactions ranging from self-limited urticaria to anaphylaxis, erythema multiforme, angioedema, pneumonia, myocarditis, Guillain–Barré syndrome, thrombocytopenia, and the aggravation or onset of autoimmune diseases [14,15,16]. Allergists and the media have suspected that these problems may have arisen due to specific vaccine ingredients, such as anaphylaxis occurring in response to polyethylene glycol (PEG), and have also considered that these reactions may have occurred due to spike protein antigen presenting, or foreign body contamination occurring during manufacturing or injection.

Since some hypersensitivity reactions occurring after COVID-19 infection are similar to those reported after COVID-19 vaccination, we speculate that aggravating factors may cause an infection-like reaction following COVID-19 vaccination. Herein, we review articles published in the scientific literature on the potential immune mechanisms regarding COVID-19-vaccination-related hypersensitivity reactions (including thrombosis and thrombocytopenia, cutaneous adverse reactions, myocarditis, perimyocarditis) to the three most widely administered COVID-19 vaccines (namely the BNT162b2, mRNA-1273, and AZD1222 vaccines).

## 2. COVID-19 Vaccine: Structure and Immune Reactions

The structure of SARS-CoV-2 and its associated immune reactions are the basis of vaccine development. Similar to other coronaviruses, in addition to +ssRNA, the viral particle of SARS-CoV-2 is composed of the spike S protein (for adherence and host cell membrane fusion), the envelope E protein (for viral production and maturation), the membrane M protein (which generates the primary structure of the envelope), and the nucleocapsid N protein (for RNA replication and immune evasion) (Figure 1a) [2,3,10]. The S protein is a trimeric envelope glycoprotein that contains two subunits. Namely, this protein contains S1 and S2 subunits for receptor binding and membrane fusion, respectively. The S1 subunit has an N-terminal domain and a receptor-binding domain (RBD), which attach to the angiotensin-converting enzyme 2 (ACE2) receptor on the surface of the host cells [2,3,10]. Upon the binding of the S1 subunit to the ACE2 receptor, the type II transmembrane serine proteases (TMPRSS2) cleave the S glycoprotein and ACE2 to activate entry and augment viral uptake (Figure 1a) [3,17].

The engagement with ACE2 receptors also explains the invasion routes, affected organs, and the tendency towards angioedema and urticaria evidenced following infection (Figure 1b,c) [18,19,20,21,22]. 

SARS-CoV and SARS-CoV-2 have similar entry receptors (i.e., ACE2 receptors). To induce an immune response against the spike S protein of SARS-CoV-2, researchers designed codon-optimized sequences for efficient expression of the full-length S protein in the mRNA vaccines (BNT162b2 of BioNTech-Pfizer and mRNA-1273 of Moderna). To enable the smooth entry of genetic sequences inside the antigen-presenting cells (APCs) without undergoing enzymatic degradation in the human body, researchers have developed several types of nanoparticles, including lipid-based (LNPs), polymeric, inorganic, virus-like, or replication-deficient adenovirus vectors. This facilitates the encapsulation of nucleic acid sequences as well as their delivery to cells (Figure 1d–f) [3]. Both vaccine constructs (BNT162b2 and mRNA-1273) include the two stabilizing mutations in S2 (K986P and V987P), and they are formulated as complexes with specific lipids in the form of lipid nanoparticles (LNP) (Figure 1d) [3,23,24,25]. DNA vaccines have also been designed as genetic sequences encoding the glycoprotein spike (S) antigen inside a replication-deficient chimpanzee adenovirus vector (i.e., in the AZD1222 vaccine) or a replication-deficient adenovirus type 26 vector (i.e., in the Johnson & Johnson/Janssen COVID-19 vaccine) (Figure 1d) [25,26].

The BNT162b2 vaccine (developed by Pfizer–BioNTech) and the mRNA-1273 vaccine (developed by Moderna) are the first mRNA vaccines in human history [27]. The mRNA can be translated into proteins in the cytoplasm without entering the nucleus (as in DNA vaccines). Subsequently, the proteins are broken down into antigenic fragments presenting on the cell membrane, thereby completing the antigen presentation sequence (Figure 1e,f) [3]. However, there was a report that the antigen presentation of spike protein fragments, which may contain water-soluble glycoprotein fragments as allergens, may also activate the allergen-specific B cells producing IgE antibodies and trigger acute hypersensitivity reactions [28].

### 2.1. Pfizer (BioNTech): BNT162b2 Vaccine

The FDA approved the Pfizer–BioNTech COVID-19 vaccine on 11 December 2020 under an emergency use authorization (EUA). This mRNA vaccine encodes the S protein of SARS-CoV-2 encapsulated by PEGylated LNP. Before authorization, the FDA reviewed an ongoing phase 1/2/3 trial and the available efficacy data from 36,523 participants older than 12 years of age without evidence of SARS-CoV-2 infection seven days prior to vaccination. This data showed 95% vaccine efficacy (95% credible interval: 90.3–97.6%) at seven days following two doses of the vaccine (0.3 mL, 30 μg) administered at an interval of 21 days [3,29].

### 2.2. Moderna: mRNA-1273 Vaccine

The FDA approved the Moderna COVID-19 vaccine on 18 December 2020 under the EUA. This mRNA vaccine encodes the S protein of SARS-CoV-2 (with two proline mutations) and is enveloped by PEGylated LNPs. Before authorization, the FDA reviewed an ongoing phase 3 trial and the available efficacy data from 28,207 participants older than 18 years of age without evidence of SARS-CoV-2 infection prior to receiving their first dose of the vaccine. The data showed 94.1% vaccine efficacy (95% credible interval: 89.3–96.8%) 14 days following the second dose of the 0.5 mL vaccine, which was administered 28 days after the first dose. Moreover, the vaccine efficacy was found to reach 94.5% after seven weeks of follow-up [3,30].

### 2.3. Oxford–AstraZeneca Vaccine: AZD1222

This DNA vaccine comprises genetic sequences encoding the glycoprotein spike (S) antigen enveloped by a replication-deficient chimpanzee adenovirus vector. It was designed as a nonprofit vaccine to ensure an equal global supply in low- and middle-income countries [31]. Phase III clinical trial results, published on 16 December 2021, based on an analysis of the data of 17,662 fully vaccinated participants, revealed that two doses of the vaccine showed an overall efficacy of 74.0% (95% credible interval: 65.3–80.5%) after a median follow-up period of 61 days. Moreover, an increase in neutralizing antibodies was seen at 28 days following the second dose of the vaccine [32].

## 3. Acute Hypersensitivity Reaction to Vaccine Excipients (PEG 2000, Polysorbate 80)

There were reports of immediate and severe reactions occurring at marginally higher frequencies than for other vaccines following the initiation of worldwide vaccination campaigns. According to the U.S. Centers for Disease Control and Prevention (CDC) and the Vaccine Adverse Event Reporting System (VAERS), 11.1 people per million doses (71% within 15 min) demonstrated anaphylaxis following the Pfizer–BioNTech COVID-19 vaccination in the U.S. between 14 and 23 December 2020. Moreover, anaphylaxis was observed in approximately five people per million COVID-19 vaccinations following one year of approval [33,34]. In contrast, the rate of anaphylaxis following other vaccinations in the period between January 2009 and December 2011 was 1.31 persons per million vaccinations in a Boston investigation [35]. However, there were no severe side effects reported in clinical trials of mRNA vaccines. This may be because some phase III trials specified “known or suspected allergy or history of anaphylaxis, urticaria, or other significant adverse reaction to the vaccine or its excipients” as an exclusion criterion [36].

The VAERS data during the period between 14 and 23 December 2020, revealed that 81% of the vaccine recipients who reported anaphylaxis had a previous history of environmental allergies or allergic reactions to foods, drugs, or pharmaceuticals, compared with 33% of recipients with a history of anaphylaxis (including two cases following rabies and influenza A (H1N1) vaccinations (one each)). In addition, 90% of such recipients comprised women [33]. The majority of those reporting acute hypersensitivity reactions (worldwide) experienced reactions within 30 min of vaccination. Moreover, the majority of these recipients presented with an allergic history to drugs or pharmaceuticals. Some allergen-specific immunoglobulin E (IgE) antibodies to the proteins, glycoproteins, or excipients found in COVID-19 vaccines may already exist in the human body [28,37,38].

Regarding vaccine excipients, BNT162b2 utilized PEG 2000, whereas mRNA-1273 utilized PEG 2000 and tromethamine as LNP materials for improving the stability of lipid nanoparticles (Figure 1d). In addition to PEG, a few vector vaccines (the ChAdOx1 nCoV-19 vaccine and the Johnson & Johnson/Janssen COVID-19 vaccine) used polysorbate 80 (P80) as a vaccine excipient [26,29,30,37]. PEGs are hydrophilic polymers of ethylene oxide that are 200 g/mol to 35,000 g/mol in molecular weight and are widely applied as solubilizers, stabilizers, or emulsifiers in pharmaceutical, medical, food, industrial, and cosmetic applications. Moreover, PEGs are potential allergens triggering immediate-type hypersensitivity. The majority of pharmaceuticals (i.e., injectable medications such as methylprednisolone acetate, medicines for constipation, or medicines for bowel preparation before colonoscopy) use PEG 3350. In addition, PEGs of different molecular weights have been used in vaccines as the stabilizing portion of the liposome (e.g., PEG 2000). Though PEG was considered to be a relatively stable product, several reports have revealed the existence of 0.1–9% anti-PEG IgG, immunoglobulin M (IgM), and/or IgE following treatment with PEGylated therapeutics [39,40].

Among other excipients with potential to cause allergies, polysorbates are derivatives of PEGs that carry the possibility of structural cross-reactions due to a shared chemical moiety (namely –(CH_2_CH_2_O)_n_). However, few supporting cases have been reported to date [28,37,38]. Moreover, polysorbates are used as excipients of other vaccines, such as influenza, hepatitis A, hepatitis B, and rotavirus vaccines [39]. In addition, tromethamine (trometamol; molecular formula: C4H11NO3), one of the excipients in mRNA-1273 vaccine, is a buffer ingredient for regulating the pH of nucleic solutions, which is also widely used in several medications or cosmetic products. It was reported as the causal ingredient in a case of contact dermatitis based on the positive result of trometamol 1% aqueous skin prick tests (SPTs) [41]. According to a recent report, it also triggered IgE-mediated trometamol allergy in a 23-year-old patient who developed anaphylaxis after receiving gadolinium-based contrast agents [42].

Researchers have proposed a hypothesis postulating that PEGs in fragments of LNPs float on the surface of APCs and interact with IgE antibodies following LNP endocytosis [28]. Moreover, the PEGs-IgE complex bound to the FcεR1 receptor on mast cells or basophils may trigger the release of histamines, prostaglandins, proteases, and other inflammatory mediators, thereby causing anaphylaxis or acute hypersensitivity reactions (Figure 1f,g) [28,38].

In addition to antibody-mediated immunity to PEGylated lipids, researchers have speculated as to the existence of various pseudoallergic responses without previous exposure, leading to the direct degranulation of mast cells or the activation of mast-cell-independent mechanisms (Figure 1g) [38]. Moreover, sex hormones are one of the factors that may influence mast cell degranulation, through which estrogen promotes or testosterone and progesterone diminish the T helper cell type 2 (Th2) response. Moreover, these hormones facilitate or suppress histamine release from mast cells, thereby partly explaining the higher percentage of anaphylaxis cases seen in women [38].

Besides the marginally higher anaphylaxis rate, an overemphasis on acute hypersensitivity reactions following vaccination leads to hesitation in regard to receiving initial or additional doses of COVID-19 vaccines for individuals with an allergic constitution. This requires physicians to use a flowchart or perform allergy tests to determine contraindications to mRNA vaccination and/or indications for alternative vaccines (Table 1). A case series report analyzing data from the Stanford Medicine Network (18 December 2020–27 January 2021) identified 148 patients who received the International Classification of Diseases 10th Revision diagnoses of anaphylaxis among 38,895 instances of vaccinations. Of these 148 cases, 82 (55%) had a history of prior COVID-19 vaccination (46 with the BNT162B2 vaccine, 29 with the mRNA-1273 vaccine, and 7 who received an unknown vaccine type), and 22 were categorized as experiencing vaccine-related allergic reactions. In addition, 17 cases met the Brighton anaphylaxis criteria (14 BNT162B2, 3 mRNA-1273). Of these 17 patients, 11 underwent SPTs, which revealed 0 positive responses to PEG, 0 positive responses to polysorbate 80, and 1 positive response to a similar brand of the mRNA vaccine. In contrast, basophil activation testing (BAT) revealed 10 positive reactions to PEG and 11 positive reactions to the administered mRNA vaccine [43]. BATs can be activated by IgG via complement activation-related pseudoallergies or by IgE via IgE-FcεRec activation. Therefore, the authors speculated that the vaccine–material immune complexes (e.g., PEG immune complexes) may be more likely to exist in the blood than in the skin and that anaphylaxis reactions may mostly occur due to a non-IgE mediated pathway (i.e., via complement activation through plasma immune complexes with the materials), according to the negative result of the administered SPTs (Figure 1g) [43].

In addition, a cohort study that reviewed data collected at the Mass General Brigham Hospital in Boston in a study conducted between 6 January 2021 and 3 March 2021 identified 65 of 80 cases of allergic reactions following the first dose of a COVID-19 vaccine; these allergic reactions occurred within 4 hours of undergoing vaccination. These researchers administered SPTs for alternative PEG 3350 products and P80 products and found that tolerance to the second dose of vaccination was not associated with the results of PEG or P80 SPTs [44]. Considering the widespread usage of PEG in several products without a concomitant high frequency of anaphylaxis, the structure of PEGylated lipid nanoparticles is speculated to be one of the causative factors.

In addition, a small study recruited three patients with histories of PEG allergies (two were caused by macrogol containing aperients, one by a parenteral steroid preparation), and one of them also had a history of polysorbate allergy. They underwent SPTs and intradermal tests (IDTs) for products of PEG 3350, P80, discarded BNT162b2, and AZD1222. All of these patients had positive IDTs for BNT162b2 (PEGylated lipid nanoparticles), and one patient with a history of polysorbate allergy also had a positive reaction to AZD1222 (with concurrent negative SPTs for Optive advanced, Cellufresh, methylprednisolone acetate, methylprednisolone succinate, triamcinolone, BNT162b2, AZD1222, and positive SPTs for Movicol). BATs were performed for PEG (200–6000 g/mol), BNT162b2, AZD1222, and PEGylated liposomal doxorubicin, revealing CD63 upregulation only in response to BNT162b2 and PEGylated liposomal doxorubicin. Therefore, basophil activation to BNT162b2 was presumably mediated by PEGylated liposomes under complement activation-related pseudoallergy rather the native state of PEG [45].

Considering that an increasing number of articles have shown positive BAT results (Table 1), a prospective study enrolled 17 patients who experienced hypersensitivity reactions following the first dose of BNT162b2; this study was conducted in Malaga, Spain, with a start date in January 2021. The enrolled patients underwent SPTs, IDTs, and BATs for PEG 1500, PEG 3350, and BNT162b2. Besides 11 non-allergy cases presenting with lighter symptoms who received a second vaccination with tolerance, the remaining positive cases were divided into a PEG allergy group (two cases with positive SPTs and/or BATs for PEG), those sensitized to the vaccine (two cases with positive SPT and/or BAT findings in regard to the vaccine), and an undetermined group (i.e., patients with negative SPTs and BATs for both PEG and the vaccine). In addition, the authors evaluated a control group without PEG allergies, comprised of five patients who recovered from COVID-19 without vaccination; five vaccinated patients who recovered from COVID-19 and who were vaccinated with BNT162b2 without experiencing an allergic reaction; four patients who had not experienced either COVID-19 infection or vaccination; and four patients who had not experienced COVID-19 infection and who were vaccinated with BNT162b2 without an allergic reaction on their BAT. A total of 50% of the control cases with a history of SARS-CoV-2 infection were positive for BNT162b2 on the BAT, whereas others in the control group had not experienced a COVID-19 infection presented with negative findings. Thus, this study determined that BATs were effective for determining PEG allergies; however, a positive result to the vaccine may indicate a history of SARS-CoV-2 infection [46].

Although DNA-based COVID-19 vaccines do not contain PEGs and can provide an alternative option for individuals with PEG allergies, people may hesitate to receive these vaccines due to the structural cross-reaction between PEGs and polysorbates. Sellaturay et al. enrolled eight patients with a history of positive SPTs or IDTs to PEGs and performed P80 and AZD1222 SPTs [47]. Of the eight participants, one patient had a positive test for only P80, and one patient tested positively for both P80 and AZD1222. These patients received two courses of AZD vaccination with tolerance under pre-dosing antihistamines with 1 hour of monitoring, confirming that those with PEG allergies can be vaccinated with AZD1222. Although two patients presented with positive SPTs to P80, the researchers speculated that there may be variations in the level of resulting influence, since the amount of P80 in AZD1222 is <100 µg (which is considerably less than in other non-COVID-19 vaccines). Moreover, the Johnson & Johnson/Janssen COVID-19 vaccine contains a lower level of P80 (0.16 mg), thus providing more choices for patients with both PEG and P80 allergies [47].

Acute hypersensitivity reactions may be induced due to excipients such as PEG, tromethamine, or polysorbate, or proteins transcribed from the genetic sequences in the administered vaccines. Moreover, allergy tests may be inconsistent with tolerance to COVID-19 vaccination. Nevertheless, these tests still have a role in evaluating individuals with hypersensitive constitutions to select a suitable vaccine under proper pre-medication or postvaccine monitoring conditions (Table 1 and Table 2).

## 4. Immunologic Mechanisms for Subacute or Chronic Adverse Reactions

Besides acute hypersensitivity reactions, researchers have reported on the development of several autoimmune-related symptoms in a few weeks following vaccinations. These reactions have occurred in a range of settings worldwide. However, there is a lack of direct evidence, as the majority of studies are case reports, case series, or retrospective data analyses. Discussions of underlying mechanisms have mostly concerned speculations such as the immunological balance of these autoimmune diseases and the theoretical reaction to a vaccine. Herein, we present a brief review of the most discussed adverse reaction of thrombosis/thrombocytopenia, cutaneous reactions, and myocarditis/perimyocarditis.

### 4.1. Thrombosis and Thrombocytopenia

An interim report from a two-year planned surveillance investigation analyzed and evaluated 23 patients experiencing serious outcomes following mRNA vaccination; these data were obtained from the Vaccine Safety Datalink between 14 December 2020 and 26 June 2021. Frequently reported serious events comprised thrombosis-related diseases, including stroke, myocardial infarction, and organ-related embolism. The incidence of serious outcomes was not statistically significantly higher in the period occurring from 1 to 21 days postvaccination, compared with 22–42 days postvaccination among individuals with similar characteristics [48].

To evaluate the incidence in the general population, a population-based cohort study of cardiovascular and hemostatic events occurring in the first 28 days following AZD1222 vaccination in people aged 18–65 years was conducted in Denmark and Norway between 9 February and 11 March 2021. This study reported 11 excess venous thromboembolism events per million vaccinations, along with 2.5 excess cerebral venous thrombosis events per million vaccinations, as compared with the number of expected events in the general population. However, the authors concluded that the absolute risks were small upon comparing the benefits of vaccination against the risks of infection [49].

In addition, a descriptive analysis evaluating thromboembolism events following the administration of different COVID-19 vaccines calculated the incidence of outcomes arising from BNT162b2, mRNA-1273, and AZD1222 vaccinations between 13 December 2020 and 16 March 2021, using data from the WHO Global Database for Individual Case Safety Reports (VigiBase, https://who-umc.org/vigibase/, accessed on 16 March 2021 ). These data were based on post-marketing surveillance using self-reports; therefore, self-reporting biases may exist as well as other commonly occurring biases in the collected data. Nonetheless, this study demonstrated that thrombotic events may occur in association with all three vaccines, with 31.8% (381/1197) vs. 67.9% (813/1197), 24.6% (80/325) vs. 77.6% (253/325), and 52.2% (334/639) vs. 48.2% (308/639) for venous thrombotic events and arterial thrombotic events, respectively, occurring following BNT162b2, mRNA-1273, and AZD1222 vaccination, respectively [50].

In addition, Scully et al. examined 23 patients with suspected vaccine-induced thrombosis and thrombocytopenia without a history of hereditary or acquired thrombophilia. In consideration of the presence of progressive thrombosis, thrombocytopenia, and symptoms similar to heparin-induced thrombocytopenia (HIT), these researchers determined the existence of antibodies to platelet factor 4 (PF4) [51]. Anti-PF4 antibody is positive in HIT and connects complexes between heparin and platelet-released PF4, contributing to the aggregation and activation of platelets and inducing thrombosis [26,51]. Despite a lack of heparin exposure, 22 patients presented with positive anti-PF4 test findings via enzyme-linked immunosorbent assay (ELISA), leading to speculation of autoinflammation triggered by vaccination or the vaccine itself. The authors termed this phenomenon pathogenic PF4-dependent syndrome, which was considered unrelated to the use of heparin therapy, and they recommended the administration of intravenous immune globulin (IVIG) or glucocorticoids [51]. The mechanisms underlying the formation of the antibody-binding PF4 complex (without heparin) are still unknown; however, some studies have hypothesized that the DNA or RNA component of the vaccine is exposed before reaching the cell due to improper preservation, thus causing the formation of PF4/aptamer complexes (Figure 1h) [26,52]. Moreover, adenovirus vectors may increase the likelihood of thrombocytopenia by activating platelets and inducing platelet-leukocyte aggregation [52,53]. Despite an unknown mechanism, “Vaccine-induced Immune Thrombotic Thrombocytopenia (VITT)” has been defined in communications from the U.S. CDC and FDA. In addition, the American Society of Hematology declared the five mandatory diagnostic criteria for this condition to be as follows: COVID-19 vaccination occurring 4–42 days prior to symptom onset, any venous or arterial thrombosis (often cerebral or abdominal), thrombocytopenia (platelet count < 150 × 10^9^/L), positive PF4 “HIT” ELISA findings, and markedly elevated D-dimer levels (> four times of the upper limit of normal) [54].

### 4.2. Cutaneous Adverse Reactions

Self-limited cutaneous reactions beyond the injection site have been restricted to case reports or case series owing to their heterogeneity. A previously published comprehensive review of cutaneous reactions analyzed data from the PubMed, EMBASE, and Web of Science databases and documented a <1% incidence of COVID-19 arm, pityriasis rosea, urticarial, chilblain-like lesions, vasculitis, erythema multiforme, Rowell’s syndrome, herpes zoster, herpes labialis, lichen planus, petechial skin rash secondary to VITT, or morbilliform rash, and derived the conclusion that there was an associated risk of transient, benign, and self-limited symptomology following vaccination. The authors also reported a higher tendency toward these reactions in patients with previous allergies and mast cell disorders [15].

In addition, a nationwide Spanish cross-sectional study analyzed patients with cutaneous reactions occurring within 21 days of vaccination (from 16 February to 15 May 2021). This study reported that 40.2% of reactions (*n* = 405 reactions) were to the BNT162b2 vaccine, 36.3% were to the mRNA-1273 vaccine, and 23.5% were to the AZD1222 vaccine. Besides the acute reaction of the COVID-19 arm (32.1%) as well as an urticarial reaction (14.6%), both of which occurred with relatively high frequency, morbilliform (8.9%; mean onset at 4.0 days) frequently developed following BNT162b2 (52.8%) and AZD1222 (30.5%) vaccination. In addition, these researchers identified the occurrence of papulovesicular eruption (6.4%; with a mean onset at 6.4 days), with no differences evidenced between the vaccines. Of these reactions, 50% of the biopsied cases evidenced purpuric reactions (4.9%); these reactions frequently developed following BNT162b2 (43.8%) and AZD1222 (56.2%) vaccination, revealing small-vessel vasculitis. Moreover, a history of SARS-CoV-2 infection did not predispose patients to cutaneous or severe reactions following vaccination. The researchers also obtained latent virus activation reports in regard to varicella zoster virus reactivation (10.1%; mean onset at 6.9 days), herpes simplex virus reactivation (3.7%), and pityriasis-rosea-like reactions (4.9%, mean onset at 6.3 days), which was related to infection with human herpesvirus 6 and 7 (HHV-6 and HHV-7); the researchers hence considered that a specific immune response against SARS-Co-V2 or the S proteins within these vaccines may decrease the cell-mediated control of latent viruses [55].

In contrast, some reports have speculated that the unknown immune effects of the mRNA vaccine itself or viral particles may trigger herpes infections [15]. Despite no reports on the incidence of other inflammatory diseases, such as psoriasis, lichen planus, or bullous pemphigoid, the authors suggested the possibility that SARS-CoV-2 infection following vaccination, indirect skin reactions owing to the host immune response to infection, or delayed hypersensitivity reactions from excipients as mediating factors [55]. This conclusion can be attributed to similarities in the patterns of reactions in connection with previously reported SARS-CoV-2 cases.

In addition, a prior review article classified four modes of vaccine-related immune mechanisms underlying autoimmune-related cutaneous reactions, which eventually increased the expression of interferon-γ (IFN-γ) according to the predominant type of cutaneous reactions, such as Th1, Th2, Th17-polarized cutaneous inflammation and granulomatous and fibrogenic reactions. The authors considered the Th1-polarized T-helper-cell-related response as a vaccine- or viral-vector-induced antiviral/antitumor response, which denotes a cellular immune pattern that incorporates CD8+ T cells and macrophages. Moreover, the key cytokines of the aforementioned type of reaction include interferon-gamma (IFN-γ), tumor necrosis factor α, interleukin 2 (IL-2), and IL-6, all of which are related to lupus erythematosus, lichen planus, maculopapular, morbilliform, vesicular rash, and erythema multiforme [56]. Th2-polarized cutaneous inflammation is also related to acute and chronic urticaria, atopic dermatitis, contact dermatitis, and injection reactions and is considered the result of prior sensitization to the components or adjuvants of a vaccine, concomitant with increased levels of IL-4, IL-5, and IL-13 as well as cutaneous eosinophilia [56].

Following vaccination, a flare-up or recent onset of psoriasis vulgaris, Sweet’s syndrome, or neutrophilic and pustular drug reactions may occur due to the vaccine-derived activation of the innate immune system, thus causing the activation of skin-resident memory T cells and eliciting a Th17/Th22 reaction. Moreover, vaccine-derived trauma or the degradation of the extracellular matrix may activate macrophages, thereby leading to the development of granulation and fibrosis reactions that may be related to morphea- or scleroderma-like symptoms [56]. The recent development or exacerbation of vasculitis events was not only reported in regard to the COVID-19 vaccine but has also infrequent been reported in reaction to the influenza virus, the hepatitis B virus, the Bacille Calmette–Guerin (BCB) vaccine, and human papillomavirus, although without clear evidence to this regard.

Vaccine-related antigens inducing the formation of an immune complex presumably deposit in and damage the endothelium of vessels, provoking leukocytoclastic vasculitis. Nevertheless, SARS-CoV-2 itself may develop the hyperactivation of the immune system through cross-reactivity and molecular mimicry between the virus and self-antigens, thereby inducing autoimmune disorders. Therefore, COVID-19 vaccines containing the genetic sequence of the S protein from the virus may carry a marginal possibility of generating the abovementioned events through an as-yet unknown mechanism [16,56].

### 4.3. Myocarditis or Perimyocarditis

In addition to the abovementioned events, there have been reports of myocarditis or perimyocarditis following vaccination, as described in the previously reported case series. More specifically, the majority of cases involved adolescents presenting with chest pain, elevated troponin levels, and/or ST-segment changes on electrocardiography within 2–14 days after vaccination. Most of the reactions were self-limited or resolved following the administration of nonsteroidal anti-inflammatory medications, although some patients with myocarditis required IVIG or corticosteroid treatment [57,58].

In addition, a retrospective cohort study analyzed data from the largest healthcare organization in Israel between 20 December 2020 and 24 May 2021 and reported 54 cases of myocarditis during the 42 days elapsing after BNT162b2 vaccination among the 2,558,421 enrolled participants (i.e., 2.13 cases per 100,000 persons). Among these cases, 41 mild cases, 12 intermediate cases, and 1 fulminant case were recognized, and the highest incidence was seen in 16–29-year-old males [59].

Another retrospective cohort study likewise analyzed data from the Israeli Ministry of Health during a similar timeframe and reported 136 myocarditis cases among approximately 5.1 million participants; 119 cases occurred following a second vaccination, with the highest frequency again seen in 16–19-year-old males. More specifically, among persons aged between 16 and 19 years, myocarditis occurred in approximately 1 of the 6637 male recipients and in 1 of the 99,853 female recipients within 21 days after the second vaccination. The researchers also performed a comparative evaluation in reference to unvaccinated persons and determined that the rate ratio 30 days after the second vaccine dose in fully vaccinated recipients was 2.35 (95% confidence interval [CI], 1.10–5.02). Therefore, the authors concluded that there was an increased incidence of myocarditis occurring after the second dose of the BNT162b2 vaccine in young males [60].

According to a review by Professor Kounis, due to the lack of routine myocardial biopsy, the underlying mechanism or pathogenesis is not clear. Of eight published biopsy reports, four cases revealed eosinophilic myocardial infiltration that indicated the possibility of hypersensitivity myocarditis [61]. Other articles suggest the involvement of vaccine adjuvants, which trigger the innate immune system and enhance signal transition as relevant to adaptive immunity, thus providing a second signal for T-cell activation. Although the BNT162b2, mRNA-1273, and AZD1222 vaccines do not contain adjuvants, the mRNA/DNA of these vaccines possess self-adjuvant properties, thereby acting both as antigens and adjuvants. For mRNA vaccines, the ssRNAs recognized from endosomal Toll-like receptors (TLR3 and TLR7) in the endosome and the components of the inflammasome of melanoma-differentiation-associated protein 5 (MDA5), retinoic-acid-inducible gene I (RIG-I), NADH dehydrogenase 2 (ND2), and *protein kinase R* (PKR), bind to ssRNA and double-stranded RNA (dsRNA) in the cytosol, resulting in the production of multiple inflammatory mediators and type I interferon (IFN) [62].

In adenovirus vector vaccination, dsDNA engages with TLR9, inducing the secretion of type I IFN [62]. Although the genetic sequences of the vaccine contain modified nucleotides to reduce binding to TLRs and immune sensors, as well as to limit the possibility of excessive systemic inflammation eliciting severe side effects, a prior study found an approximately 100-fold increase in the frequency of myeloid cell clusters, higher CD14^+^, CD16^+^ inflammatory monocyte counts, higher levels of plasma IFN-γ and IFN-response transcription factors, and enhancement of the innate immune system following booster vaccination [63]. The amplified release of type I IFN also amplifies T-cell memory and B-cell differentiation and survival, thereby continuing the memory response. The Th1 response also triggers IFN-γ expression and activates cytotoxic lymphocytes and natural killer (NK) cells, which can lead to excessive cytotoxicity [63,64].

In addition to the adjuvant effects of mRNA and DNA, LNPs may have inflammatory properties. In order to define the inflammatory properties of LNPs, a study administrated empty LNPs formulated in phosphate-buffered saline (PBS) to mice through intramuscular administration found statistically significant upregulation of gene transcripts associated with the activation of inflammasomes (i.e., IL-1β and NLR pyrin domain containing 3 [NLRP3] inflammasomes), as well as downregulation of the inflammasome inhibitors of NLR10 [65]. A murine model demonstrated that the upregulation of the NLRP3 inflammasome, IL-1α, and IL-1β may exacerbate pericardial effusion and thickness, thus indicating their possible associations with pericarditis [66].

In addition to studies using animal models, a previous case report characterized the immune condition of a young man with myopericarditis following vaccination with the mRNA-1273 vaccine via a multiplex cytokine assay, a flow cytometry analysis, and an endomyocardial biopsy. The test results revealed markedly increased IL-18, IL-27, and Th1-type cytokine levels and activated circulating NK cells and T-cells. The monocytes also expressed increased levels of IL-18 and the NLRP3 inflammasome, similar to findings observed in mice with cardiac dysfunction following the administration of recombinant-IL18 [67].

To assess the incidence of myocarditis and perimyocarditis following the administration of other (non-COVID-19) vaccines, a study analyzed VAERS data from 1990 to 2021 and divided the cases into a COVID-19 vaccine group and a non-COVID-19 vaccine group. A total of 1972 myocarditis events (348 in the non-COVID-19 vaccine group and 1579 in the COVID-19 vaccine group), and 1438 pericarditis events (375 in the non-COVID-19 vaccine group and 1063 in the COVID-19 vaccine group) were reported, with 18–29-year-old males as the predominantly affected demographic group. The authors concluded that postvaccination myocarditis and perimyocarditis are not unique to COVID-19 vaccines and attributed the relatively high frequency of myocarditis and perimyocarditis associated with COVID-19 vaccination to the large number of COVID-19 vaccinations administered to date [64]. Since postvaccination myocarditis and perimyocarditis events are related to IFN-γ and the Th1 immune response, some researchers have suggested increasing the time interval between the first and booster vaccination in order to avoid overactive immune responses [63,64].

## 5. Conclusions

The development and implementation of COVID-19 vaccinations has occurred within 2 years. This is an unusually fast pace of events, adopted due to the urgent need presented by the pandemic. Hypersensitivity reactions reported in the media have elicited excessive concern. However, the majority of incidents have been self-limited. In addition to excipient-related reactions and antigens created by the translation of the mRNA or DNA in the vaccines, the nucleic acids themselves may have adjuvant effects that may trigger an overactive immune response. For people with a hypersensitive constitution or those exhibiting hypersensitivity reactions following the first dose of a vaccine, their healthcare providers can check their history of PEG allergies and perform SPTs or BATs to evaluate the need for shifting to other COVID-19 vaccines, under supervision, for the second vaccine dose. To avoid overactive immune responses from booster vaccinations, increasing the time interval between vaccine doses may be considered.

## Figures and Tables

**Figure 1 biomedicines-10-01641-f001:**
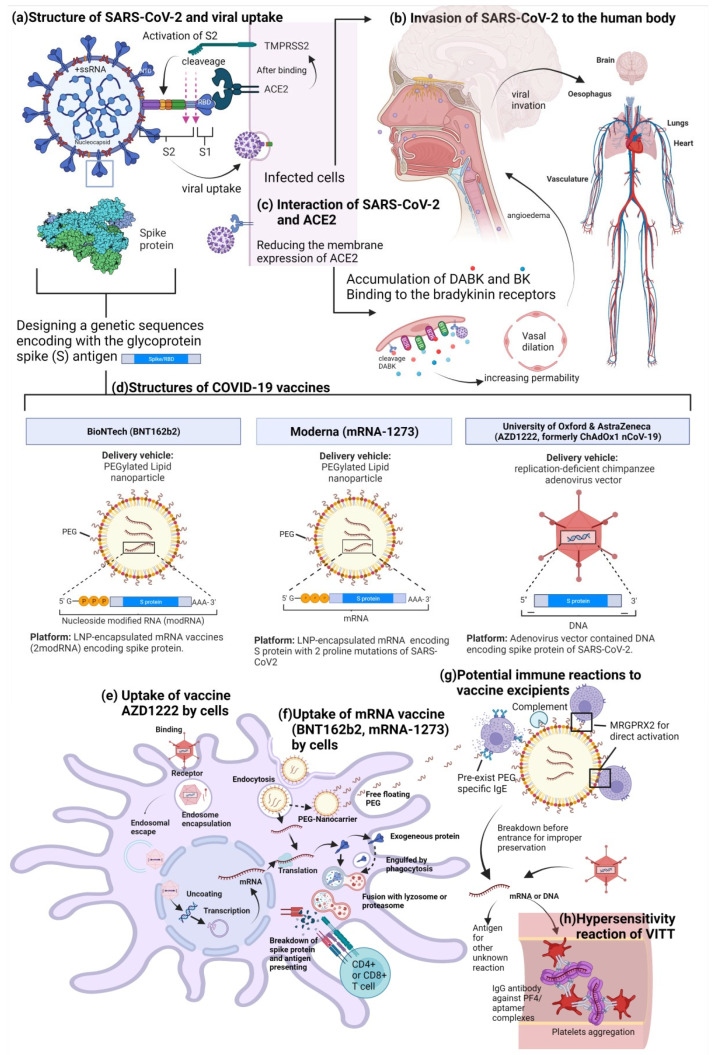
Immune response and vaccines used to fight coronavirus disease 2019 (COVID-19) and vaccination-related hypersensitivity reactions. (**a**) The structure of severe acute respiratory syndrome coronavirus 2 (SARS-CoV-2). The S protein is a trimeric envelope glycoprotein containing two subunits. Following the attachment of the receptor-binding domain (RBD) of the S1 subunit to the angiotensin converting enzyme 2 (ACE2) receptor on the host cells, type II transmembrane serine proteases (TMPRSS2) cleave the S glycoprotein and ACE2 to activate viral entry and to facilitate viral uptake. (**b**) SARS-CoV-2 invades the host through the epithelial cells of the oral mucosa, airway epithelium, or nasal cavity and targets organs and tissues expressing ACE2 receptors. (**c**) SARS-CoV-2 depletes ACE2 receptors on the endothelial cell membranes, causing the accumulation of des-Arg(9)-bradykinin (DABK) and bradykinin (BK), contributing to vasodilation, increasing vascular permeability, and triggering angioedema. (**d**) The structure of a COVID-19 vaccine containing genetic sequences encoding the glycoprotein spike antigen or the RBD. (**e**,**f**) Uptake of vaccines (AZD1222, BNT162b2, mRNA-1273) by cells. Following the endocytosis of lipid-based nanoparticles (LNPs), the free-floating polyethylene glycol (PEG) may be released to the extracellular environment. (**g**) Potential immune reactions to vaccine excipients. The vaccine excipients, such as PEG, may interact with preexisting specific immunoglobulin E (IgE) antibodies. PEG-specific IgEs may already be present in patients with an allergic history to PEG. PEGs-IgE complex bound to the FcεR1 receptor on mast cells or basophils trigger the release of histamines, prostaglandins, proteases, and other inflammatory mediators, causing anaphylaxis or acute hypersensitivity reactions. The hypothesized mechanism of the action of pseudoallergens involves the activation of mast cells through the direct stimulation of Mas-related G-protein-coupled receptor member X2 (MRGPRX2) by PEGs, or complement activation without preexisting anti-PEG antibodies, thereby leading to a nonclassical anaphylactic response. (**h**) Hypersensitivity reaction of vaccine-induced immune thrombotic thrombocytopenia (VITT). The vaccines’ components may interact with IgG antibodies to form platelet factor 4 (PF4)/aptamer complexes. These inter-actions may contribute to the aggregation and activation of platelets and the induction of thrombosis. This figure was created using BioRender.com, last accessed on 27 June 2022.

**Table 1 biomedicines-10-01641-t001:** Acute hypersensitivity reactions following the first dose of coronavirus disease 2019 (COVID-19) vaccination, as well as results of skin prick tests, intradermal tests, and/or basophil activation tests.

Study (Month and Year)	Study Period and Sample Size	Examined Drug	Results	Conclusions
Warren et al., April 2021 [43]	18 December 2020–27 January 2021 Stanford Medicine Network 14 cases receiving the BNT162b2 vaccine 3 cases receiving the mRNA-1273 vaccine (91% women)	**SPT:** DMG-PEG 2000 (Avanti Polar Lipids, 1 μg/μL) P80 (Millipore Sigma; Sigma Aldrich, 1 μg/μL) Undiluted remnant vaccine PC: Histamine (1 mg/mL) NC: Filtered saline **BAT:** DMG-PEG 2000 (same as SPT) P80 (same as SPT) Vaccine remnant (0.007 μg/μL) PC: Anti-IgE (Bethyl Laboratories; 1 μg/mL) NC: Filtered saline	11/17 received examinations **SPT:** 0 patients were positive to PEG 0 patients were positive to P80 1 patient was positive to the vaccine **BAT:** 10 patients were positive to PEG 11 patients were positive to the vaccine Anti-PEG IgG was detectable in positive reactions in the vaccine group Anti-PEG IgE was not detectable in any of the groups	Most cases may be attributed to non-IgE-mediated immune responses to PEG
Wolfson et al., June 2021 [44]	6 January 2021 to 3 March 2021 Mass General Brigham 80 allergic reactions 65 reacitons occurred within 4 h, whereas 15 reactions occurred within 4 h to three days 18 patietns received the BNT162b2 vaccine 62 patients received the mRNA-1273 vaccine (89% women)	**SPT:** 1:1, 1:10, 1:100 PEG-3350 (MiraLAX) P80 (Refresh Tears)	33 patients received PEG testing only, 47 patietns received PEG and P80 testing **SPT:** Five patients tested positively to PEG (two refused the second dose) Nine patients tested positively to P80 (two refused the second dose) 66 patients tested negatively to the vaccine (six refused the second dose) **70 patients received a second dose** PEG positive: No reaction P80 positive: Three reactions SPT negative: 15 reactions	Skin test did not add information regarding the tolerance of the second dose of the mRNA COVID-19 vaccine.
Troelnikov et al., May 2021 [45]	Recruited 3 patients with a history of PEG allergy; 1 patient also presented with a polysorbate allergy 3 healthy controls	**SPT, IDT, and BAT:** Products of PEG 3350 Products of P80 BNT162b2 remnant AZD1222 remnant (PEGylated liposomal doxorubicin, only tested in a BAT)	**SPT:** Except Movicol, positive findings for 1 patient; all other tests were negative in every patient **IDT:** All patients tested positively to the BNT162b2 vaccine **BAT:** All patients tested positively to the BNT162b2 vaccine and to PEGylated liposomal doxorubicin	Basophil activation is mediated by PEGylated liposomes due to complement-activation-related pseudoallergies
Labella et al., September 2021 [46]	Regional University Hospital of Málaga January 2021 17 patients with hypersensitivity reactions following the first dose of the BNT162b2 vaccine	**SPT:** BNT162b2 as is PEG1500 (Roxall) 0.1%, 1%, and 10% PEG 3350 (Movicol) 55 mg/mL **IDT:** (if SPT negative) PEG1500 (Roxall) 0.01% PEG3350 (Movicol) 0.55 mg/mL, 5.5 mg/mL **BAT:** PEG 2000 (Sigma, St Louis, MO, USA) 100, 10, 1 and 0.1 μg/mL BNT162b2 vaccine at 10 μg/mL, 1 μg/mL, 0.1 μg/mL, and 0.01 μg/mL	**SPT:** 1 patient was positive to PEG 5 patients were negative to PEG; 4 declined receiving a second dose, and 1 received the Janssen vaccine **BAT:** PEG allergy: 2 patients were positive to PEG and BNT162b2 Vaccine sensitized: 2 patients were positive to BNT162b2 Undifferentiated: Both negative Randomized control group: 5/10 patients with a SARS-CoV-2 infection history tested positively for the BNT162-b2 vaccine	The BAT is a potential tool for determining allergies to PEG excipients, but not to the overall COVID-19 vaccine A positive result for the vaccine on the BAT indicates a past COVID-19 history
Sellaturay et al., October 2021 [47]	Cambridge University Hospital 8 patients with a history of positive SPT or IDT findings for PEG between October 2013 and February 2021	**SPT:** P80 (20%) AZD1222 (< 100 µg P80/dose) All patients received an AZD1222 vaccine following SPT, and both P80 and AZD1222-positive patients received the split dose	**SPT:** 1 patient tested positively to P80 1 patient tested positively to both P80 and AZD1222 **Direct AZD1222 vaccination:** 1 patient tolerated AZD1222 (while exhibiting anaphylaxis to BNT162b2) 6 patients tolerated 2 doses of AZD1222 1 patient with both P80 and AZD1222 SPTs positive tolerated AZD1222 (the first dose: 1/5 dosage following 30 m of observation, 4/5 dosage; the second dose: 1 dosage)	Patients with PEG allergies can be vaccinated with AZD1222, despite positive SPT findings to P80

BAT—basophil activation test; COVID-19—coronavirus disease 2019; IDT—intradermal test; NC—negative control; P80—Polysorbate 80; PC—positive control; PEG—polyethylene glycol; SPT—skin prick test.

**Table 2 biomedicines-10-01641-t002:** Brief Summary of BNT162b2, mRNA-1273, and AZD1222, as well as the associated allergy tests.

	BNT162b2	mRNA-1273	AZD1222
	mRNA Vaccine		DNA Vaccine
Active component	mRNA encoding the S glycoprotein of severe acute respiratory syndrome coronavirus 2 (SARS-CoV-2)	mRNA encoding the S glycoprotein with two proline mutations ins SARS-CoV-2	DNA encoding the glycoprotein spike (S) antigen of SARS-CoV-2
Vaccine efficacy [29,30,32]	95% (7 days following 2 vaccine doses with a 21-day interval)	94.1% (14 days following 2 vaccine doses with a 28-day interval)	74% (2 vaccine doses after a median follow-up of 61 days)
Excipients [26,29,30] Potential allergic triggers: PEG2000, tromethamine, trometamol, polysorbate 80	PEGylated LNP ALC-0315 (4-hydroxybutyl)azanediyl)bis(hexane-6,1-diyl)bis(2-hexyldecanoate) ALC-0519 2[(polyethylene glycol)-2000]-*N*,*N*-ditetradecylacetamide 1,2-distearoyl-sn-glycero-3phosphocholine Cholesterol Potassium chloride Monobasic potassium phosphate Sodium chloride Dibasic sodium phosphate dihydrate Sucrose Diluent (0.9% sodium chloride injection)	PEGylated LNP SM-102 1,2-dimyristoyl-rac-glycero-3-methoxypolyethylene glycol-2000 [PEG2000-DMG] Cholesterol 1,2-distearoyl-sn-glycero-3-phosphocholine [DSPC]) Tromethamine Tromethamine hydrochloride Acetic acid Sodium acetate Sucrose	Replication-deficient chimpanzee adenovirus vector ChAdOx1 L-histidine L-histidine hydrochloride monohydrate Magnesium chloride hexahydrate Polysorbate 80 Ethanol Sucrose Sodium chloride Disodium edetate dihydrate (EDTA) Water for injections
Acute allergic risk groups [39]	Potential anaphylaxis history to a first dose of the mRNA COVID-19 vaccine, or to injectable pharmaceuticals/vaccines or oral intake products containing PEG, PEG derivates, or polysorbates (structural cross-reactions): e.g., methylprednisolone acetate, methoxy polyethylene glycol-epoetin beta, pegfilgrastim, medroxyprogesterone acetate, Brilliant Blue G Ophthalmic Solution, sulfur hexafluoride, bimatoprost implant, trastuzumab, rilonacept, perflutren lipid microspheres, PEGylated liposomal doxorubicin, and other medications (totaling more than 1000 FDA approved medications)		Same as the mRNA vaccines. Vaccines containing polysorbate: influenza, hepatitis A/B, Tdap, HPV, DTaP/DtaP + IPV/DtaP + HepB + IPV/DtaP + IPV + HepB + Hib, Japanese encephalitis, pneumococcal 13-valent, rotavirus, zoster, meningococcal group B, and others Other pharmaceuticals: antiarrhythmic, antidiabetic, antifungal, anti-inflammatory, cancer treatment, vitamins, and other medications or supplements
Prevention/treatment of acute allergy [39,43,44,45,46,47]	Potential anaphylaxis history to food, oral or injected forms of pharmaceuticals without PEG, or polysorbates: → Vaccination with 30 min of observation Anaphylaxis to the first dose of the vaccine, past history of meeting the criteria for the acute allergic risk group, or an anaphylaxis history to pharmaceuticals without known ingredients (containing or not containing PEGs/polysorbates) → Referral to an allergist for clinical phenotyping and evaluating indications for skin prick tests or basophil activation tests. If the result is negative, consideration of the indication for intradermal testing		
	Under well-informed consent: Negative to all tests → Pre-dosing with antihistamines and administering vaccinations under monitoring with 30–60 min of postvaccination observation Positive to skin prick tests (SPTs/basophil activation tests (BATs), or intradermal tests (IDTs) → Consideration of a DNA vaccine (AZD1222 or the Johnson & Johnson/Janssen COVID-19 vaccine) under conditions of pre-dosing with antihistamines with 60 min of postvaccination observation [47]		Under well informed consent: Negative to all the tests → Same as the mRNA vaccines. Positive findings on SPTs/BATs or IDTs → Theoretically, consider vaccination with mRNA vaccines. However, this decision requires more clinical tests.

BATs—basophil activation tests; COVID-19—coronavirus disease 2019; DNA—deoxyribonucleic acid; DTaP—diphtheria and tetanus toxoids and acellular pertussis vaccine; EDTA—ethylenediaminetetraacetic acid; FDA—Food and Drug Administration; HepB—Hepatitis B Vaccine; Hib—*Haemophilus influenzae* type b; HPV—Human Papillomavirus; IDTs—intradermal tests; IPV—inactivated poliovirus vaccine; LNP—lipid-based nanoparticles; mRNA—messenger ribonucleic acid; PEG—polyethylene glycol; SARS-CoV-2—severe acute respiratory syndrome coronavirus 2; SPTs—skin prick tests; Tdap—tetanus, diphtheria, and acellular pertussis vaccine.

## Data Availability

Not applicable.

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
