# Peer review of "Immune Response in Regard to Hypersensitivity Reactions after COVID-19 Vaccination"

_biomedicines, 2022, doi:10.3390/biomedicines10071641_

Round 1

Reviewer 1 Report

While this article does provide a somewhat general overview of the topic, I believe that it  can be vastly improved.

1) an additional table containing relevant information about the respective vaccines would be helpful

2) the title is confusing and could be improved

3) "medical products" I am unsure if this is the correct term

4) spell-check required (e.g. SARS-CoV-2 should be uniformly spelled)

5) there is an abundance of literature about VITT, which can be a deadly side-effect, yet this is barely covered in this article. why?

6) two recent articles from the NEJM are not included in the myocarditis section.

7) rhabdomyolysis was not covered at all. why not?

8) some passages in this article are written in an incoherent manner. i suggest a major revision and maybe consider it for publication in "Vaccines", another MDPI journal

Author Response

Responses to Reviewer 1

Comments and Suggestions for Authors

While this article does provide a somewhat general overview of the topic, I believe that it  can be vastly improved.

Questions to the authors.

  • an additional table containing relevant information about the respective vaccines would be helpful

Response

Thank you for your valuable suggestion. We have added a table (Table 2) providing information on the BNT162b2, mRNA-1273, and AZD1222 vaccines, including information on allergic reactions. Your suggestion has greatly improved our manuscript. 

  • the title is confusing and could be improved

Response

Thank you for your valuable suggestion. Considering our manuscript concerns a discussion of frequent hypersensitivity reactions after vaccination, we have simplified our title as follows: “Immune Response in regard to Hypersensitivity Reactions after COVID-19 Vaccination.”

  • "medical products" I am unsure if this is the correct term

Response

Thank you for pointing out this discrepancy in terminology. We have changed this term to “pharmaceuticals” accordingly.

  • spell-check required (e.g. SARS-CoV-2 should be uniformly spelled)

Response

Thank you for your advice. We have already corrected the terms referencing “SARS-CoV-2” and have performed a spell check.

  • there is an abundance of literature about VITT, which can be a deadly side-effect, yet this is barely covered in this article. why?

Response

Thank you for your thoughtful suggestion. The main purpose of our manuscript was to discuss the possible immune mechanisms mediating hypersensitivity reactions that occur with a high frequency. Thus, instead of describing the worst outcome, we have placed our emphasis on one possible mechanism. To let readers fully understand the life-threatening implications of reactions to VITT, we have also added a brief description of deadly events that are considered to be related to VITT, using the following wording:

“Herein, we discuss the most frequently seen reactions (namely, thrombosis and thrombocytopenia), which were considered as inducing factors for ischemic stroke, brain hemorrhage, pulmonary embolism, and myocardial infarction events after vaccination. We also discuss the occurrence of myocarditis/perimyocarditis following vaccination, which has been found to occur at a relatively high frequency according to a two-year surveillance study [22, 55]. We also briefly discuss hypersensitivity reactions in regard to cutaneous reactions specifically, since the skin is the most visible organ in the human body.”

  • two recent articles from the NEJM are not included in the myocarditis section.

Response

Thank you for this valuable advice. We have added new references accordingly (reference 67 and 68) and included an additional description (lines 581-598, page 16), as follows:

“In addition, a retrospective cohort study analyzed data from the largest healthcare organization in Israel between December 20, 2020, and May 24, 2021, and reported 54 cases of myocarditis during the 42 days elapsing after BNT162b2 vaccination among the 2,558,421 enrolled participants (i.e., 2.13 cases per 100,000 persons). Among these cases, 41 mild cases, 12 intermediate cases, and one fulminant case were recognized, and the highest incidence was seen in 16- to 29-year-old males [67].

Another retrospective cohort study likewise analyzed data from the Israeli Ministry of Health during a similar timeframe and reported 136 myocarditis cases among approximately 5.1 million participants; 119 cases occurred following a second vaccination, with the highest frequency again seen in 16- to 19-year old males. More specifically, among persons aged between 16 and 19 years, myocarditis occurred in approximately 1 of the 6,637 male recipients and in 1 of the 99,853 female recipients within 21 days after the second vaccination. The researchers also performed a comparative evaluation in reference to unvaccinated persons and determined that the rate ratio 30 days after the second vaccine dose in fully vaccinated recipients was 2.35 (95% confidence interval [CI], 1.10 to 5.02). Therefore, the authors concluded that there was an increased incidence of myocarditis occurring after the second dose of the BNT162b2 vaccine in young males [68].”

  • rhabdomyolysis was not covered at all. why not?

Response

Thank you for your question. There were many hypersensitivity reactions reported after COVID-19 vaccination in every research area. However, it was difficult to discuss all the mechanisms mediating hypersensitivity reactions within the scope of this review. Therefore, we chose to discuss reactions presenting with relatively high frequencies based on two years of surveillance data described in reference 55; this reference did not mention rhabdomyolysis events. To inform readers as to why we specifically choose VITT, cutaneous events, and myocarditis/perimyocarditis in the discussion of subacute/chronic reactions, we have added an additional description as follows:

“Herein, we discuss the most frequently seen reactions (namely, thrombosis and thrombocytopenia), which were considered as inducing factors for ischemic stroke, brain hemorrhage, pulmonary embolism, and myocardial infarction events after vaccination. We also discuss the occurrence of myocarditis/perimyocarditis following vaccination, which has been found to occur at a relatively high frequency according to a two-year surveillance study [22, 55]. We also briefly discuss hypersensitivity reactions in regard to cutaneous reactions specifically, since the skin is the most visible organ in the human body.” (line 435-442 of page 13)

8) some passages in this article are written in an incoherent manner. i suggest a major revision and maybe consider it for publication in "Vaccines", another MDPI journal

Response

Thank you for your thoughtful question. The decision to write this manuscript was reached because we received an invitation for the special issue “Hypersensitivity to Drugs and Vaccines: Molecular Basis and Translational Research” from the editors of your esteemed journal. Considering the journal’s focus area (i.e., biomedicines), we decided to provide a brief overview of possible mechanisms mediating the most emphasized reactions. We chose to discuss the structure of the vaccine and respective immune reactions associated with this vaccine in the introduction because we believe it is basic to understand the main structure and reactions for the discussed vaccines before we can explain why these reactions occurred. In addition, the structural component of this review is only presented in the introductory portion of the manuscript, whereas the other components of the review are mainly devoted to discussing the biological aspects underlying the described hypersensitivity mechanisms. To let the readers fully understand why the hypersensitivity reactions occurred following COVID-19 vaccination, we designed our manuscript in a systemic manner, as follows: (a) COVID-19 vaccine: structure and immune reactions; (b) acute hypersensitivity reaction to vaccine excipients (polyethylene glycol [PEG] 2000, polysorbate 80, and skin tests); (c) angiotensin-converting enzyme 2 (ACE2) receptor and angioedema; and (d) possible immunologic mechanisms mediating subacute or chronic hypersensitivity reactions, including thrombosis and thrombocytopenia, cutaneous adverse reactions, and myocarditis or perimyocarditis. We think that our manuscript is suitable for Biomedicines and thank you in advance for your time and consideration on this invited submission.

Reviewer 2 Report

In this review, Hsieh and Yamaguchi discussed the effects of COVID-19 vaccine structure, excipients, and the immune response on hypersensitivity reactions 
after COVID-19 vaccination. The review includes the following points: a) . COVID-19 vaccine, structure, and immune reactions.   b) Acute hypersensitivity reaction to vaccine excipients (PEG 2000, Polysorbate80)
skin test.  c) ACE2 receptor and angioedema. d)Possible immunologic mechanism of subacute or chronic hypersensitivity reactions: including thrombosis and thrombocytopenia, cutaneous adverse reactions, and myocarditis or perimyocarditis and other autoimmune diseases.

In general some missing information should be included:

a) Kounis syndrome asscoaited with COVID-19 . And what is the difference between hypersentivity and Kounis syndrome.

b) Treatment and prevention strategies for allergic reactions associated with CoviD vaccine.

c) The part of vaccine excipients should be expanded to verify the following points: 1) difference excipients present in different vaccines    2)What are the main components affecting the allergic reactions    3) risk groups reported with allergic reactions.    

d) What are differences between allergic reaction after second dose vs first doses in different vaccines

Author Response

Responses to Reviewer 2

Comments and Suggestions for Authors

In this review, Hsieh and Yamaguchi discussed the effects of COVID-19 vaccine structure, excipients, and the immune response on hypersensitivity reactions

after COVID-19 vaccination. The review includes the following points: a) . COVID-19 vaccine, structure, and immune reactions.   b) Acute hypersensitivity reaction to vaccine excipients (PEG 2000, Polysorbate80)

skin test.  c) ACE2 receptor and angioedema. d)Possible immunologic mechanism of subacute or chronic hypersensitivity reactions: including thrombosis and thrombocytopenia, cutaneous adverse reactions, and myocarditis or perimyocarditis and other autoimmune diseases.

In general some missing information should be included:

  1. a) Kounis syndrome asscoaited with COVID-19. And what is the difference between hypersentivity and Kounis syndrome.

Response

Thank you for your thoughtful question. According to a past review of Kounis syndrome presenting in the absence of COVID-19, this syndrome is considered to present as vasospastic allergic angina or allergic myocardial infarction due to the eosinophil and mast cell reactions/infiltration that induce vasospasm or occluding thrombus. This could happen as an acute response. We note that this condition does not have a high frequency in male adolescents. Theoretically, this reaction could happen after the COVD-19 vaccination, but most such reactions have been limited to case reports. With regard to myocarditis, according to a recent brief review published by Professor Kounis (see https://doi.org/10.1159/000524224), the pathogenesis of these cardiac events is poorly understood because of the lack of routine myocardial biopsy. Until now, of only eight biopsy reports published worldwide, four cases demonstrated eosinophilic myocardial infiltration (which is similar to hypersensitivity myocarditis). However, there was no direct evidence allowing the researchers to determine whether the eosinophilic infiltration was due to the vaccine or its excipients. Professor Kounis concluded that the vaccine could be offered to most people. However, for people with a history of allergies to polyethylene glycol (PEG) or polysorbates, there may be a need to wait for new vaccines containing different excipients (which may or may not be difficult to develop in the future). Because this field of research and development contains many uncertainties and may lead to necessary fear of vaccination, we did not directly address the reviewer’s question in our manuscript.

  1. b) Treatment and prevention strategies for allergic reactions associated with CoviD vaccine.

Response

Thank you for your suggestion. We have already discussed treatment and prevention strategies in the original manuscript (see lines 317 to 330 of page 8, 9). To let the readers comprehend this information more easily, we have also added a new table (Table 2) to this revision.   

  1. c) The part of vaccine excipients should be expanded to verify the following points: 1) difference excipients present in different vaccines 2)What are the main components affecting the allergic reactions 3) risk groups reported with allergic reactions.  

Response

Thank you for your thoughtful advice. We had already described these factors in part 3 of the original version of the manuscript. In consideration of your advice, we have provided a summary of these points in Table 2. Thank you for your suggestion, which has made our manuscript clearer and more accessible to a wider readership.

  1. d) What are differences between allergic reaction after second dose vs first doses in different vaccines 

Response

Thank you for your question. As reported in reference 41, people experiencing anaphylaxis due to PEG or their first dose of an mRNA vaccine may consider shifting to AZD1222 under supervision in order to achieve sufficient vaccine efficacy without anaphylaxis. Before COVID-19 vaccination, previous studies have demonstrated great success with heterologous vaccination by evoking both cellular and humoral responses, resulting in a much (4-10 times) higher T-cell response. In addition, many studies have revealed an increase in IgG antibodies, neutralizing antibodies, and cellular immune responses given the first dose of AZD1222 followed by a boost with BNT162b2, and these increased responses were achieved without severe adverse effects (see https://doi.org/10.1002/jmv.27463). However, a report from Nature also indicated that the intervals between heterologous doses were longer than those seen with homologous vaccination; these researchers also mentioned “combining two different vaccines, both of which might have their own profile of adverse events and effects, which could amplify any problems.” Most studies were conducted on only approximately a few hundred people, and it was difficult to observe rare events due to insufficient statistical power (see https://www.nature.com/articles/d41586-021-01805-2). In conclusion, due to the above uncertainties, it is difficult to say with absolute certainty whether the evaluated vaccines are safe or unsafe or to compare differences in allergic reactions following heterologous and homologous vaccination.

Round 2

Reviewer 1 Report

no further comments

Author Response

Thank you for the careful review

Reviewer 2 Report

The authors addressed my questions.

Author Response

Thank you for the careful review